# MASON: Scalable, Contiguous Sequencing
# for Building Consistent Services

Christopher Hodsdon[*†], Theano Stavrinos[*], Ethan Katz-Bassett[†], Wyatt Lloyd[*]

[*]*Princeton University,* [†]*Columbia University*

## Abstract

Some recent services use a sequencer to simplify ordering operations on sharded data. The sequencer assigns each operation a multi-sequence number which explicitly orders the operation on each shard it accesses. Existing sequencers have two shortcomings. First, failures can result in some multi-sequence numbers never being assigned, exposing a noncontiguous multi-sequence, which requires complex scaffolding to handle. Second, existing implementations use single-machine sequencers, limiting service throughput to the ordering throughput of one machine.

We make two contributions. First, we posit that sequencers should expose our new contiguous multi-sequence abstraction. Contiguity guarantees every sequence number is assigned an operation, simplifying the abstraction. Second, we design and implement MASON, the first system to expose the contiguous multi-sequence abstraction and the first to provide a scalable multi-sequence. MASON is thus an ideal building block for consistent, scalable services. Our evaluation shows MASON unlocks scalable throughput for two strongly-consistent services built on it.

## 1  Introduction

Designers of large-scale distributed services grapple with the tradeoff between strong consistency on one hand and high performance on the other hand. A strongly-consistent distributed service is a useful building block because applications can reason about its behavior as if it were running on a single machine. However, strong consistency requires coordination among a service's servers, adding overhead.

Some recent services achieve consistency using a *sequencer* to explicitly order data accesses a priori, removing the need to coordinate concurrent accesses [37, 56]. This enables sequencer-based designs to achieve strong consistency with higher throughput than other approaches.

Early work using sequencers used a *sequence* abstraction that globally orders all operations [3, 49]. More recent work, and this work, target the *multi-sequence* abstraction that only explicitly orders operations that execute on the same subset of data [37, 56]. This allows operations spanning multiple subsets of data to only be ordered with respect to other operations on intersecting subsets, reducing contention compared to ordering all operations globally, and improving throughput and latency.

The multi-sequence abstraction uses a collection of *sequence spaces*, i.e., logically independent sequences of strictly increasing integers, to provide a strictly serializable ordering of accesses to different subsets (*shards*) of the service's data. An operation that needs cross-shard ordering gets an atomically assigned *multi-sequence number* containing a sequence number from the sequence space of each shard the operation accesses. An *execution protocol*, designed by the service developer, defines the sequence spaces involved in an operation and how shards use multi-sequence numbers to execute operations. Driven by the execution protocol, the service's servers use the sequence numbers to order operations on the shard(s) they manage, with the multi-sequence numbers atomically ordering operations relative to other operations to provide strong consistency. Operations ordered by multi-sequence numbers can be executed without coordination across servers, enabling strongly consistent, scalable, and efficient services.

However, the abstraction used by recent services is a *noncontiguous multi-sequence*: failures can cause *holes* in the sequence space, i.e., sequence numbers that are never used. To preserve consistency, a service must identify and reason about all holes. Identifying holes requires service-wide coordination between the service's servers to reach consensus on whether a sequence number has an associated operation that can be recovered. If not, then it is a hole, and the servers must coordinate to avoid using any sequence numbers that are part of the same multi-sequence number as the hole. Implementing consensus and service-wide coordination to handle holes significantly complicates execution protocol design (§2.2).

This paper introduces the *contiguous multi-sequence* abstraction for building consistent, scalable services. The contiguous multi-sequence abstraction assigns exactly one operation to every integer in each sequence space such that no sequence space has a hole. Contiguity strengthens the multi-sequence abstraction over its existing noncontiguous counterpart by hiding consensus and service-wide coordina-

tion, simplifying the development of services. Some existing services use the noncontiguous multi-sequence abstraction internally to expose higher-level abstractions like distributed databases [37, 56]. Compared to higher-level abstractions, the contiguous multi-sequence supports developing more diverse functionality, e.g., ephemeral objects (§6).

In addition to being noncontiguous, existing implementations of the multi-sequence abstraction [37, 56] suffer from a second limitation: they have an ordering throughput ceiling that limits the throughput of any services built on top of them. These implementations use a *monolithic sequencer*, a single machine whose only task is to hand out multi-sequence numbers, enabling low-latency ordering that is easy to reason about. A monolithic sequencer can order operations with higher throughput than coordination-based mechanisms, but this design can *only achieve ordering throughput up to the throughput limit of a single machine.* Thus, a service built on a monolithic sequencer cannot scale.

Our system, MASON, addresses the ordering throughput limitation. MASON is a building block for distributed services that provides the contiguous multi-sequence abstraction with no ceiling on ordering throughput, unlocking scalability for services that were previously unscalable. MASON's contiguous multi-sequence implementation enables services to (1) use simple execution protocols that need not incorporate consensus or service-wide coordination and (2) scale to achieve service throughput far higher than what is possible with monolithic sequencers.

Our key insight is that MASON can enable simple execution protocols and scalability via a layer of replicated proxies between clients, which send operations, and a monolithic sequencer. To overcome the failure modes that expose holes, the proxy layer provides fault tolerance for clients and the sequencer. The proxy layer replicates enough of each client operation's state to ensure the operation can be completed and a sequencer can recover without holes, thus guaranteeing the contiguous multi-sequence abstraction.

To overcome the monolithic sequencer's ceiling on ordering throughput, proxies batch requests for multi-sequence numbers. This batching is perfect, in that the sequencer does no more work to allocate one million contiguous numbers than it does to allocate a single number. Each replicated proxy operates essentially independently, allowing the proxy layer to scale out; adding more proxies increases ordering throughput. These techniques enable MASON to scale: if the sequencer is the bottleneck, proxies increase batch size; if the proxy layer is the bottleneck, more proxies are added.

Our evaluation shows MASON provides scalable ordering throughput: with one sequence space, MASON achieves ~16.7 Mops/sec with 24 proxy machines, scaling to ~31.5 Mops/sec with 48 proxy machines. MASON's tradeoff for a stronger abstraction and scalable ordering throughput is higher latency relative to monolithic-sequencer designs, since the proxies and a single round of replication are on path for each request. MASON's latency is still low, however, with a median latency of ~243 $\mu$s at the reported throughputs.

We demonstrate MASON's value as a building block by using it to implement Corfu-MASON, a distributed shared log modeled after CORFU [3]; and ZK-MASON, a distributed prototype of the coordination service ZooKeeper [22]. With MASON's strong abstraction, it was easy to build these services that consistently execute cross-shard operations (§6). MASON also unlocked scalability for them in contrast to their fundamentally unscalable original designs. Specifically, our implementation of CORFU's original design is limited to ~14.1 Mops/s (nearly line rate for a sequencer with a 10G NIC, ~14.5 Mops/s). Building it on MASON lets it scale from ~7.3 Mops/s (one server) to ~29.1 Mops/s (four servers). Our implementation of ZooKeeper's original design is limited to ~150 Kops/s; its MASON-based implementation scales from ~1.3 Mops/s (one server) to ~7 Mops/s (eight servers).

This paper makes two major contributions. The first is the contiguous multi-sequence abstraction, which simplifies building correct services compared to the previous noncontiguous multi-sequence abstraction. While the noncontiguous multi-sequence abstraction demands significant distributed systems expertise to use correctly, our abstraction shields service developers from the complexity of reasoning about holes (§2). By handling this complexity internally, the contiguous multi-sequence abstraction enables faster development of new services, promotes designs with fewer bugs, and enables developers without distributed systems expertise to develop scalable distributed services. The second major contribution is the design of MASON, which notably is the first multi-sequence design that is scalable. MASON's inherent scalability is the foundation for removing the throughput ceiling from existing and future services built on a multi-sequence abstraction (§5). Together, these contributions make it easy to build consistent services with a newfound ability to scale service throughput (§6).

## 2 The Contiguous Multi-Sequence

This section is an orientation to the multi-sequence abstraction. Section 2.1 explains how to build strongly-consistent services with the generic multi-sequence abstraction. Section 2.2 describes why building services with the existing *noncontiguous* multi-sequence abstraction is challenging. Our *contiguous* multi-sequence abstraction instead makes it easy to use multi-sequences to build scalable, consistent services.

### 2.1 Building Services with Multi-Sequences

The sequence abstraction globally orders operations in a single sequence space. The multi-sequence abstraction extends the sequence abstraction to multiple sequence spaces to enable the service to order operations only when they execute on the same subset of data. This enables services to execute operations in order with less coordination: servers managing a subset of data only need local ordering information,

reducing contention compared to ordering all operations globally, improving throughput and latency. Services built on the generic multi-sequence abstraction typically include clients, a sequencing component, and servers, each holding one or more shards. Typically, each shard stores a subset of the service's data and is replicated for fault tolerance. Each shard has its own sequence space, a sequence of strictly increasing integers that order operations on the shard's data. To execute an operation, a client identifies the shards involved in the operation, gets a multi-sequence number from the sequencing component with one number from each relevant shard's sequence space, and sends the operation to the shards' servers with the multi-sequence number. Each server locally uses the multi-sequence number to order this operation's data accesses relative to other operations' accesses. In contrast, with a single, global sequence space, each shard would need to know the next operation to execute across all shards instead of only the next operation concerning its own data.

We next define multi-sequence numbers, explain how they are assigned to operations consistently, and describe how execution protocols use them to scale execution.

**Multi-sequence numbers.** A *multi-sequence number*, $n$, is a set of $\langle ssid, sn \rangle$ tuples where *ssid* is a unique number identifying the sequence space, and *sn* is a sequence number in that space. The sequence number in space $s$ in multi-sequence number $n$ is denoted $n_s$. For a set of sequence spaces requested by a client, the sequencing component returns a multi-sequence number consisting of the next sequence number $n_s$ in each relevant space $s$.

**Strictly serializable multi-sequence number assignment.** From clients' perspectives, strictly serializable services process operations one at a time in an order that a single machine could have received them [48]. Concretely, *strict serializability* requires that there exists a legal total order of operations consistent with the partial ordering of "real-time" precedence, i.e., if $a$ completes before $b$ begins, then $a$ must be ordered before $b$ [21, 48].

Multi-sequence numbers enable strongly consistent distributed services when assigned to operations in a strictly serializable order. To simplify discussion, we define a default, $\Delta$, where $n_s = \Delta$ for all $n_s$ not mapped to a specific sequence number (i.e., all $s$ not in this multi-sequence number). For the set of all sequence spaces $S$, we define a partial ordering over all multi-sequence numbers where $a < b \iff \forall s \in S, a_s \neq \Delta \wedge b_s \neq \Delta \implies a_s < b_s$. The multi-sequence abstraction guarantees that two multi-sequence numbers either share no common sequence spaces or are strictly ordered (i.e., if $a_s < b_s$ for one common space $s$, then $a_{s'} < b_{s'}$ for all common spaces $s'$, implying $a < b$). The partial ordering of the multi-sequence numbers defines the ordering of operations. If strict serializability imposes an ordering between two operations, then multi-sequence numbers assigned on path with their execution capture that ordering.

**Execution protocols.** To use the multi-sequence abstraction, a service developer implements an *execution protocol* that executes operations in order of their multi-sequence numbers, yielding a strictly serializable service. The execution protocol runs on clients (typically encapsulated in a client library) and on the service's servers. For clients, the execution protocol defines how operations are mapped to the service's shards and which sequence spaces are involved in a given operation. For servers, it determines when shards can safely execute operations, based on the operations' multi-sequence numbers.

**Scalable execution.** Multi-sequence numbers enable services to scale throughput up to the rate the sequencer can assign sequence numbers. Execution scales through parallelism: when some shards are executing an operation, other shards can execute a different operation. The sequence spaces in multi-sequence numbers determine which operations can execute in parallel, as operations with disjoint multi-sequence numbers access different shards. As long as multi-sequence number assignment keeps up, the service can increase its throughput by adding more machines and creating more shards. However, existing multi-sequenced services use *monolithic* (single-machine) sequencers, which can never assign sequence numbers to operations at a higher rate than a single machine can support and hence limit the service's scalability.

## 2.2 From Noncontiguous to Contiguous

The generic multi-sequence abstraction is realized as a *noncontiguous* abstraction in existing services, which use it to expose higher-level abstractions [37, 56]. As we explain next, noncontiguity complicates service development. In contrast, the *contiguous* multi-sequence abstraction simplifies developing services with multi-sequences by encapsulating that complexity within the abstraction.

**Holes in a noncontiguous sequence complicate the abstraction.** *Holes* occur when a sequence number is not used for an operation. For example, a hole occurs if a client fails after receiving a sequence number but before using it. A shard may see, e.g., sequence numbers 1–3 and then receive an operation with sequence number 5, indicating a potential hole at 4. To preserve strict serializability, the shard may only execute operation 5 after 4 is used, since 4 could belong to any operation. To make progress in the absence of an operation, the service must decide that the entire multi-sequence number is a hole and enforce that it is not used on any shard, typically by assigning a *no-op* to each of its sequence numbers.

Handling holes complicates service design. The service must have a mechanism to identify sequence numbers that are potential holes. Existing designs use timeouts [3, 56] or infer holes from out-of-order operation arrival [37, 56]. More challenging is that the service's servers must reach service-wide consensus on whether a sequence number is a hole, then coordinate to ensure that the other numbers in the hole's multi-sequence number are treated as holes to avoid partially executing a cross-shard operation. Existing services achieve

this with a global shared log [56] or a failure coordinator [37]. Requiring consensus in the execution protocol makes a service developer's task significantly more difficult. Consensus is hard to implement and incorporate [7, 47], and requires developers to understand the nuances of the sequencing component and consensus implementation in depth.

Although existing services feature workable solutions for handling holes, requiring services to select and properly incorporate a solution does not reflect operational best practices. Much of the purpose of providing infrastructure building blocks (such as an implementation of the multi-sequence abstraction) is to enable services to use them without needing to understand their complexities, via clean abstractions that mask the subtleties of their internal operation and failure modes. Pushing the complexity of handling holes to services increases the chances of one doing so incorrectly, similar to how pushing memory management to individual programmers increases the chances of memory leaks.

**Our contiguous multi-sequence avoids holes and hides consensus.** Our abstraction assigns exactly one operation to each sequence number in each sequence space. Service developers can focus on designing execution protocols that achieve their services' goals, a much simpler task when freed from reasoning about holes or implementing consensus. Eris [37] and vCorfu [56], the two existing designs built on the noncontiguous multi-sequence abstraction, were developed by distributed systems experts. With the contiguous multi-sequence abstraction, we aim to empower developers without such expertise to use multi-sequences to build scalable, consistent services, and make it easier and faster for experts.

## 3 MASON Overview

The central contributions of MASON are to shield services from the complexity of dealing with holes by providing the contiguous multi-sequence, and to provide the benefits of the multi-sequence abstraction while allowing ordering throughput to scale beyond what a monolithic sequencer can provide. Section 4 describes how the components work together to guarantee a contiguous multi-sequence. Section 5 describes how MASON enables scalability with two mechanisms that relieve all throughput bottlenecks.

### 3.1 Model and Assumptions

We assume a set of processes that communicate via point-to-point communication over an asynchronous network, where messages can be arbitrarily delayed and reordered. We assume a crash failure model, where processes execute according to their specification until they cease sending messages and the failure is undetectable to other processes. MASON is safe under these assumptions. We assume service shards implement at-most-once semantics to handle retransmissions.

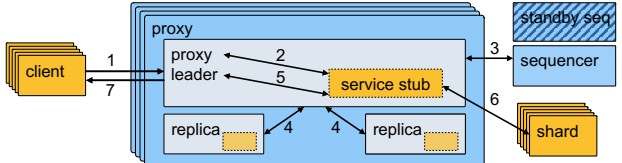

**Figure 1: The components of a service built with MASON and an operation's flow through the service. Blue components are part of MASON; yellow components are supplied by the service. Numbers correspond to steps in §3.3.**

### 3.2 MASON Components

Figure 1 shows how MASON is used in a service. It also shows MASON's two types of internal components: a *sequencer* and replicated *proxies*. The core of MASON's design is a monolithic sequencer that provides high-throughput operation ordering, surrounded by a replicated proxy layer that handles the failure modes and bottlenecks impeding existing sequencers.

**The sequencer** allocates increasing multi-sequence numbers. It is implemented by a single machine, and only one sequencer is active at a time. MASON maintains a cold *standby sequencer* for failure recovery. The standby sequencer does not participate in normal operation; it is only required for liveness. The monolithic sequencer at MASON's core provides the benefits of existing sequencers: contention-free, high-throughput ordering of operations in a distributed system. In our system, MASON itself is the distributed system, leveraging the sequencer's benefits while managing its drawbacks to provide a simpler, scalable building block to the service.

**Proxies** are replicated state machines (RSMs). We treat the RSM protocol as a black box, but for liveness require that it is leader-based and informs a replica when it gains and loses leadership. These two properties enable a leader takeover protocol that allows MASON to only replicate state after receiving the response from the sequencer, discussed below (§4.1). Both requirements could be eschewed at the cost of adding normal-case latency by replicating state both before making a request to the sequencer and after receiving a response. However, because leader-based protocols are common in practice, we choose the lower latency design. Our implementation uses Raft [46]. A proxy is logically a single entity implemented by a leader process and multiple follower processes on separate machines. The leader accepts operations from clients and executes them via the *service stub* using multi-sequence numbers. The rest of this paper refers to a proxy replica group simply as a *proxy*. A MASON deployment may have one or more proxies, depending on system load.

Identical to many other RSM-based systems, we assume at most $f$ of $2f + 1$ proxy replicas fail [30, 36, 41, 46]. A MASON deployment must be configured so that $f$ is sufficiently large. In the rare event that more than $f$ machines fail, manual intervention by an operator is necessary to restore availability.

**Service stubs** are implemented by the service built on MA-

SON and drive the execution protocol on the proxies. Service developers interact with MASON on the proxies through service stubs which execute within the proxy's process. When a proxy receives an operation from a service's client, it passes the operation to the stub. The stub either requests that MASON order the operation, or executes the operation immediately if it need not be ordered, e.g., an inconsistent read. After ordering and replicating the operation, the proxy returns it back to the stub which begins the execution protocol. Stubs are analogous to client libraries in existing multi-sequenced services. Section 6 shows how stubs are used to develop services.

The proxy may *batch* requests for multi-sequence numbers for scalability, i.e., request multi-sequence numbers for multiple client operations in a single sequencer request (§5). The sequencer *allocates* a multi-sequence number for each operation in the batch. An allocated multi-sequence number is one given to a proxy that the sequencer promises not to allocate again. Proxies *assign* multi-sequence numbers to client operations. Assignment uses replication to permanently associate a multi-sequence number with an operation and guarantee it will never be assigned to another operation. Once the proxy has replicated the assignment of a multi-sequence number to an operation, it returns the operation and its multi-sequence number to the service stub for execution.

### 3.3  Normal-Case Operation of MASON

The normal case operation of MASON, shown in Figure 1, includes the following steps:

1. A client sends an operation to a proxy.
2. The proxy passes the operation to the service stub which determines the relevant sequence spaces.
3. The proxy asks the sequencer to allocate a multi-sequence number covering the relevant sequence spaces.
4. The proxy replicates the allocated number and operation, assigning the number to the operation.
5. The proxy returns the operation and multi-sequence number to the service stub.
6. The service stub and shards run the execution protocol.
7. The proxy sends the response from the stub to the client.

## 4  Ensuring a Contiguous Multi-Sequence

MASON provides a contiguous multi-sequence by handling all potential sources of holes: client failures (Figure 2a), network drops (Figure 2b), sequencer failures (Figure 2c), and combinations thereof. This section covers how MASON handles each of these failure scenarios and then sketches a proof of strict serializability.

### 4.1  Proxies Prevent Holes from Client Failure

In a multi-sequenced service, client failure can cause holes when the client obtains a sequence number and fails before it uses the sequence number in the service. For instance, Figure 2a shows Client A failing before using sequence number

3 from sequence space $i$, resulting in a hole at 3. MASON prevents such holes with proxies that manage multi-sequence numbers on clients' behalf. Proxies are replicated for fault tolerance, eliminating this source of holes. A proxy will always return an operation that was assigned a multi-sequence number to the service stub even if the client fails and even if a minority of proxy replicas fails.

A byproduct of replication is that proxies maintain a record of every assigned sequence number, which is used in sequencer recovery (§4.3). By masking client failure and maintaining state needed for sequencer recovery, proxy replication is a key mechanism for avoiding holes in MASON.

The proxy replication strategy is driven by correctness and performance. Proxies must replicate enough information to preserve contiguity and strict serializability. Replicating every input to the proxy leader would be correct, but this would add unacceptable latency to client requests and burden proxies with excessive communication overhead. Fortunately, MASON can skip replication for all but one step in operation processing, because the other steps can be safely retried, including after client, sequencer, and/or proxy replica failure.

The exception is step 5 (Fig. 1), returning a multi-sequenced client operation to the service stub. Replicating the mapping of each client operation to a multi-sequence number before this step is critical for correctness in MASON. Suppose the mapping is not replicated. The sequencer and proxy leader could fail concurrently after the leader returns a multi-sequenced client operation to its service stub, but before the stub sends its operation to every relevant shard. The shards that received the operation may execute it, but the operation will not be completed after recovery because the mapping of multi-sequence number to operation was lost. Exposing the partial execution violates strict serializability. Therefore, before returning an operation to the service stub, the proxy must permanently associate the operation with a multi-sequence number through replication. Once replication succeeds, the sequence number is *assigned* to the operation.

We next describe how the proxy processes operations, in order to explain why all other steps are safe to retry. We discuss one operation and a single sequence space for ease of explanation; the reasoning can be easily extended to batches of operations and multiple sequence spaces.

**Receiving a client operation.** Clients can send an operation to any proxy. When a proxy leader receives an operation from a client, it passes the operation to the service stub. If the stub requests that the operation be ordered, the leader allocates a *sequencer request ID* for that operation (step 3 in Figure 1). Sequencer request IDs are allocated only by the leader, so they are trivially contiguous and strictly increasing. Sequencer request IDs are used during proxy failover to recover sequence numbers that were allocated but not yet assigned to any operation, i.e., potential holes.

**Requesting a sequence number.** The leader then requests a sequence number from the sequencer, with the sequencer re-

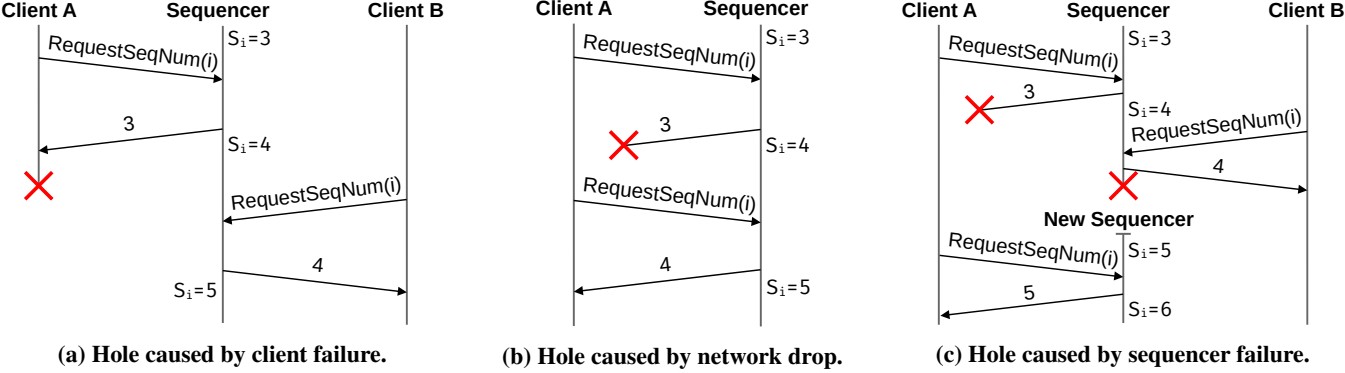

**Figure 2: Potential sources of holes.** $S_i$ **is the next sequence number in sequence space** $i$**.**

quest ID (step 3). If the sequencer has not seen this sequencer request ID from this proxy, the sequencer updates its state in two relevant ways: it allocates a sequence number for this request by incrementing the sequence counter in the requested sequence space, and maps the sequencer request ID to the allocated number. If the sequencer has seen the sequencer request ID before, it responds with the previously allocated sequence number and marks it as a retransmit.

**Proxy leader failure.** When a proxy leader fails, the new leader must recover the sequence numbers that were allocated but not yet assigned. The state needed to correctly match allocated but unassigned sequence numbers to operations was lost with the failed leader, so these are temporary holes. We now explain how we use sequencer request IDs to recover such holes. This is the key mechanism for ensuring correctness when proxies execute only one round of replication.

The new leader collaborates with the sequencer to identify these temporary holes as follows:

1. The new leader saw a contiguous set of sequencer request IDs until some ID $x$, after which it saw noncontiguous IDs until $y$. The range from $x$ to $y$ is noncontiguous because the leader replicates sequence number-operation pairs as they arrive from the sequencer, which may be out of order.
2. The new leader requests sequence numbers for all IDs from $x+1$ until $y-1$ that were not replicated. The sequencer will either return already-allocated sequence numbers, or will allocate new numbers for the IDs.
3. The new leader replicates and assigns all returned sequence numbers to no-ops and returns them to the service stub.
4. The new leader then resumes normal operation, allocating sequencer request IDs from $y+1$.

There may be allocated but unassigned sequence numbers with sequence request IDs greater than $y$. In such cases, the sequencer will mark the returned sequence numbers as retransmits. The leader replicates and assigns them to no-ops and retries the request with a new sequencer request ID. If the sequencer fails concurrently with leader failure, the sequencer recovery protocol recovers and assigns no-ops to any

allocated but unassigned sequence numbers (§4.3).

**Returning the operation and sequence number to the service stub (step 5).** Strict serializability dictates that the service's execution protocol cannot use one sequence number for multiple operations, and different sequence numbers cannot be used for one operation. MASON must therefore guarantee the sequence number associated with an operation never changes once the service is made aware of it. MASON thus replicates the sequence number-to-operation assignment (step 4) before passing the operation to the service stub.

The proxy leader's other steps in handling a client operation—passing the operation to the service stub and forwarding the service's response to the client (step 7)—can be safely left unreplicated. Retrying these steps is safe. The service stub, shards, and clients already provide at-most-once semantics to handle retransmission due to network drops, so they will be able to handle retransmission from the proxies.

## 4.2 Reliable Transport Prevents Holes from Packet Loss

Holes can occur after network drops (as illustrated in Figure 2b). MASON handles network drops with a reliable transport layer. Since the state needed to reliably transport multisequence numbers is lost on sequencer failure, MASON uses a recovery protocol to correctly fill holes with no-ops in case of simultaneous packet loss and sequencer failure (§4.3). Reliable transport and the sequencer recovery protocol ensure that every allocated multi-sequence number arrives at a proxy.

## 4.3 Recovering to Prevent Holes from Sequencer Failure

Sequencer failure can cause holes if failure occurs before the reliable transport protocol can retransmit a dropped response. For instance, suppose the sequencer allocates and sends the sequence number 3 for sequence space $i$ and later 4 for the same sequence space $i$ for two client operations, as illustrated in Figure 2c. If the message containing 3 is dropped and the sequencer fails before retransmission, but a client receives 4, then 3 is a temporary hole. One solution replicates

the sequencer to permanently associate client requests and multi-sequence numbers. However, replication compromises the main benefit of a sequencer: simplified ordering so the sequencer can devote all its resources to allocating numbers.

MASON instead runs one active sequencer, backed by an idle standby sequencer and sequencer recovery protocol. If the active sequencer fails, the standby sequencer takes over and executes the recovery protocol to correctly fill any temporary holes caused by the failure, ensuring a contiguous multi-sequence when the standby resumes normal operation.

MASON's sequencer recovery protocol is based on two observations. First, the proxies' collective state includes which sequence numbers have been assigned, so they collectively know where potential holes in each sequence space are. MASON assigns these sequence numbers to no-ops. Second, all outstanding operations are concurrent. An outstanding operation is one that a proxy received (step 1 in Fig. 1), but has not yet assigned a sequence number (step 4), and thus is not ordered. When the standby sequencer resumes normal operation, it can allocate new multi-sequence numbers for outstanding operations in any relative order, as long as they are ordered after the highest previously-assigned sequence number in each sequence space, which the proxies collectively know.

The steps in MASON's sequencer recovery protocol are:

*a)* Detect sequencer failure and activate a standby sequencer.
*b)* Identify potential holes in each sequence space.
*c)* Replicate the assignment of no-ops to holes.
*d)* Resume normal operation with new sequence numbers.

**Failure detection and standby sequencer activation.** Proxies unreliably detect sequencer failure with timeouts and pings. If a proxy does not hear from the sequencer after a timeout (.5 s in our implementation), it pings the sequencer. After another timeout, the proxy declares the sequencer failed and initiates recovery by activating the standby sequencer. The standby sequencer informs the other proxies that recovery has begun. All proxies then replicate a special recovery operation and seal their sequence spaces, rejecting any packets from the previous sequencer. The new sequencer waits for all proxies to complete the sealing process before resuming recovery. Replicating the recovery operation on all proxies before allowing the standby sequencer to resume recovery ensures proxies reject all packets from the previous sequencer. This in turn, ensures there is only one active sequencer at a time even when proxy leaders fail, sequencer-failure detection is incorrect, or messages from the previous sequencer were delayed or reordered in the network.

**Identifying potential holes.** During normal operation, proxies track their *local views* of each sequence space. A proxy's local view is the subsequence of numbers in each sequence space that the proxy has assigned to operations. After sealing, proxies send their local views to the standby sequencer. The

standby sequencer reconstructs each sequence space, exposing any temporary holes. Garbage collection of proxies' local views is described at the end of this section.

**Assigning temporary holes to no-ops.** The standby sequencer notifies proxies of any temporary holes in each sequence space. Proxies assign these sequence numbers to no-ops, replicate the assignment, and pass them to the service stubs, as they would with client-issued operations.

**Resuming normal operation.** The standby sequencer identifies the start of each sequence space based on the highest number in each sequence space compiled from the proxies. It then notifies proxies to resume normal operation and allocates new sequence numbers from that point. Proxies must re-request sequence numbers for all outstanding operations.

**Concurrent proxy leader failure.** If a proxy leader fails during sequencer recovery, the new leader will have enough state to correctly participate in recovery. In particular, if the leader fails before replicating the recovery operation, the sequencer will not have started recovery, and all holes normally recovered as part of leader recovery will be found when the sequencer is rebuilding the sequence. If the leader fails after replicating the recovery operation, the new leader will have the same state as the previous leader when the previous leader began sequencer recovery.

**Garbage-collecting sequence number tracking state.** Proxies run a lightweight garbage collection protocol to discard tracked sequence numbers that are no longer needed for sequencer recovery. Each sequence space is partitioned into intervals of size $N$. When all $N$ sequence numbers in an interval have been assigned to operations, it is safe to discard the state associated with those sequence numbers. To determine when all $N$ numbers have been assigned, the proxies form a communication ring and periodically send an accumulating count of the sequence numbers assigned in each sequence space's latest interval. At the end of a round, if any sequence space's count is $N$, the interval is completely assigned; all state associated with that interval is discarded.

## 4.4 Proof Sketch of Strict Serializability

This subsection sketches a proof of the strict serializability of the assignment of multi-sequence numbers to operations. The formal proof is in §A. We make the assumptions stated in §3.1. Our proof reasons about pairs of operations, showing they are either *strictly concurrent*, where they do not share sequence spaces, or *strictly ordered*, where if $a_n < b_n$ for some overlapping sequence space $n$, then $a_{n'} < b_{n'}$ for all overlapping sequence spaces $n'$, where $a_n$ denotes the sequence number in sequence space $n$ assigned to operation $a$.

To show that there exists a total order over all completed operations consistent with the partial ordering of real-time precedence, we exhaustively analyzed all cases of failure scenarios from no failures to concurrent failure of proxy leaders, proxy followers, and sequencer. In all cases an operation is assigned

at most one multi-sequence number which occurs if/when replication to a majority of replicas in a proxy succeeds. The assigned multi-sequence numbers for all operations that access overlapping sequence spaces are then strictly ordered by either the same sequencer, or by an initial sequencer and a standby sequencer that recovers all previous assignments before allocating any new multi-sequence numbers. Thus, the partial order of assigned multi-sequence numbers strictly orders all conflicting operations. Further, this partial order is consistent with real-time precedence either trivially when two operations are ordered by the same sequencer or because a standby sequencer only allocates numbers larger than the maximum previously assigned in each sequence space. Only strictly concurrent (i.e., no overlapping sequence spaces) operations are unordered by that partial order, and any ordering of them results in a valid total order. Extending the partial order to a total order consistent with real-time precedence is thus trivial: unordered operations are first ordered by the partial order of real-time precedence and then remaining unordered operations are arbitrarily ordered.

## 5 Supporting Scalable Throughput

A service's achievable throughput (*service throughput*) is capped by the minimum of the rate at which it can execute requests (*execution throughput*) and the rate at which it can order requests (*ordering throughput*). Execution throughput scales when more service shards are added if and only if the service implements a scalable execution protocol. Ordering throughput scales only if the ordering component scales. Previous multi-sequence abstraction designs do not scale.

MASON supports scalable service throughput by removing the bottlenecks that limit monolithic-sequencer designs and achieving scalable ordering. This section describes two complementary mechanisms that alleviate all ordering throughput bottlenecks: horizontally scaling out the proxy layer, and batching requests to the sequencer.

**Potential ordering throughput bottlenecks.** MASON has two components, so there are two potential bottlenecks on computation: the proxy layer and the sequencer. Each component sends and receives network traffic, so there are four potential bottlenecks on network bandwidth. Our two scaling mechanisms address all six bottlenecks: scaling out the proxy layer relieves all bottlenecks at the proxy layer, and batching relieves all bottlenecks at the sequencer.

**The proxy layer scales out.** When MASON is bottlenecked by a proxy layer resource, the proxy layer can scale out. Each proxy operates essentially independently, so holding all else constant, doubling the number of proxies doubles the amount of computation and bandwidth available at the proxies for processing client operations, doubling the proxy layer's achievable throughput.

In truth, proxies are not completely independent; there is overhead to garbage collect multi-sequence number tracking state (§4.3). However, the overhead is constant for each proxy with respect to the number of proxies because of the ring communication pattern; thus, it does not affect the proxy layer's scalability.

In §3–4 we have assumed a static configuration where the numbers of proxies and shards do not change. MASON components can be reconfigured as follows. To add a new proxy, the new proxy first creates a connection to the sequencer and then joins the garbage collection ring using standard techniques, e.g., those used in distributed hash tables [52]. Removing proxies is more difficult to do safely. For example, if a proxy is removed and the sequencer fails, the recovery protocol may not be able to reconstruct a complete view of the used sequence numbers (i.e., it will be missing those numbers used by the removed proxy but which were not yet garbage-collected). It may attempt to assign those used sequence numbers to no-ops, which is not safe. Thus, to remove a proxy, the proxy stops processing client requests, but continues to take part in garbage collection until all sequence numbers the proxy received are garbage collected. At this point the proxy can remove itself from the ring and disconnect from the sequencer. Waiting until all of its numbers are garbage collected ensures any used multi-sequence number will not be assigned a no-op. Alternatively, the proxy could transfer all of its sequence numbers to a different proxy, e.g., the next proxy in the garbage collection ring, and then leave the ring. Reconfiguring the service's shards can be achieved through operations internal to the service and via the service stubs.

**Batches are as efficient as single requests.** When MASON is bottlenecked by the sequencer proxies can increase throughput by batching multi-sequence number requests. This batching is perfect, holding all else constant, in that a request for one client operation uses the same resources as a request for multiple operations.

To request multi-sequence numbers for a batch of client requests, the proxy constructs a sequencer request which indicates the relevant sequence spaces and how many numbers are required from each sequence space to order the operations in the batch and sends a single sequencer request for the batch. The sequencer allocates the requested count of sequence numbers in each sequence space and replies with the lowest allocated number in each sequence space. Finally, the proxy iterates through client operations in the order they were received and gives each operation the next lowest sequence number in each of its sequence spaces.

MASON alleviates all bottlenecks on the sequencer by increasing the batch size. MASON's batching is timeout-driven: all client requests that arrive at a proxy within the timeout are batched together. By doubling the timeout (hence batch size) at a given client load, proxies can halve the rate at which they issue sequencer requests. The sequencer, in turn, would need half the resources to handle the same client load. The sequencer can thus handle twice the ordering throughput before hitting the same bottleneck. Timeout-driven batching is

naturally dynamic: higher client load results in larger batches.

**Why not batch at clients?** A strawman design for increasing ordering throughput is to batch requests at clients, which has two limitations. First, the maximum throughput is limited by the number of parallel requests a client will individually make. Second, batching at clients requires waiting until the client has issued those requests, which can substantially increase latency. In contrast, MASON's proxies can batch across any number of clients, achieving the large batches that allow it to scale. In general, naïvely adding only a batching layer to prior designs does not work, as it introduces new failure modes (e.g., batching machine failure) that require a comprehensive service redesign such as that of MASON.

# 6   Services

This section explains how services can easily use MASON and its contiguous multi-sequence abstraction to scale service throughput. We describe two services we implemented over MASON: a distributed shared log based on CORFU [3] and a distributed prototype of the coordination service ZooKeeper [22].

## 6.1   Interaction with MASON

A service's execution protocol consists of (at least) two components: *shards* and *service stubs*. Shards are implemented entirely by the service and interact with service stubs and other service-implemented components. Service stubs are the mechanism by which services interact with proxies. They determine an operation's relevant sequence spaces and request ordering via MASON if necessary, drive the execution protocol interacting with other service components, and have control of the operation until informing MASON that the operation is complete. This is sufficient for the services we implement here; more complex services may need multi-round sequencing for some operations, e.g., where the write set depends on the read set. In that case, MASON could be augmented so that the stub could request another round of ordering and include metadata, which MASON replicates and the service can use to resume execution if the current proxy leader fails.

## 6.2   Making CORFU Scalable: Corfu-MASON

CORFU is a shared log supporting append and read operations that consistently execute across shards [3]. Appends write a value to the current tail of the log. Reads return the value written to a specified log position. Many applications can be implemented with shared logs, e.g., producer-consumer queues and logging [24, 51].

We use MASON to implement Corfu-MASON, a service based on CORFU. CORFU's original implementation does not scale; although CORFU has a scalable execution protocol, the implementation is limited by the ordering throughput of its monolithic sequencer [3, 56]. By replacing the sequencer with MASON, MASON's scalable ordering combines with CORFU's scalable execution protocol to enable the whole service to scale.

Corfu-MASON uses CORFU's scalable execution protocol. The shared log is represented by a single sequence space. Appends acquire a sequence number that directly determines which log position to write. A round-robin mapping of log position-to-shard ensures append load is uniform on shards, enabling appends to execute in parallel [3].

Corfu-MASON implements two of CORFU's three operations, `append(b)` and `read(l)`. `append(b)` appends the entry *b* to the log and returns the log position *l* to which it was written. `read(l)` returns the entry at log position *l*, or an error code if the entry does not exist. CORFU implements a third operation, `fill(l)`, to fill holes in the sequence (and the log) caused by failed clients. CORFU clients detect holes in the log with a timeout and execute `fill(l)` to fill the *l*th position with junk. The timeout-and-`fill(l)` procedure is unnecessary in Corfu-MASON because of MASON's contiguous sequence.

Corfu-MASON's execution protocol uses sequence numbers for `append`s to determine which log positions to write, which in turn map to specific shards. In addition to eliminating the need for `fill` operations, MASON's contiguous sequence simplifies `read`s. If a client attempts to read a log position that has not been written yet, it can simply keep checking that log position. The contiguous sequence guarantees that the entry will eventually be written. `read`s need not be ordered and hence are not ordered or replicated by MASON; the service stub executes reads immediately. CORFU tolerates shard failure using client-driven chain replication [55], and so Corfu-MASON uses service stub-driven chain replication.

Corfu-MASON was implemented in a single day thanks to both the simplicity of CORFU and the strong abstraction of a contiguous sequence provided by MASON.

## 6.3   Making ZooKeeper Scalable: ZK-MASON

ZK-MASON is a ZooKeeper-like coordination service built on MASON. ZooKeeper [22] is a widely-used coordination service implemented on ZooKeeper Atomic Broadcast (ZAB) [25], a version of state machine replication (SMR). ZAB, like other SMR protocols, cannot scale: it is fundamentally limited by the rate a single machine can execute requests. Furthermore, ZooKeeper uses a single replicated state machine to ensure consistency, so an instance cannot be sharded. We designed ZK-MASON to be scalable, using the cross-shard consistency and scalable ordering provided by MASON.

**ZK-MASON operations.** Similar to ZooKeeper, ZK-MASON maintains a set of *znodes*. Each znode has a pathname beginning with "/" (similar to a filesystem) and data associated with it. We implemented seven operations in ZK-MASON:

- `create(`*path*`, `*data*`, `*flags*`)`: creates a znode with pathname *path* and data *data*. *flags* allows the client to specify a persistent or ephemeral znode.

- setData(*path*,*data*,*version*) : sets the data at *path* if *version* matches the current version, or if *version* is −1.
- getData(*path*,*watch*) : gets the data at *path*.
- exists(*path*,*watch*) : checks if the znode exists.
- delete(*path*,*version*) : deletes znode specified by *path* if *version* matches the current version, or if *version* is −1.
- getChildren(*path*,*watch*) : returns the children of *path*

The read operations getData, exists, and getChildren return the znode's current version. Read operations have a *watch* flag, which sets a *watch* on the znode if the flag is set. ZK-MASON watches have the same semantics as ZooKeeper watches. Watches are triggered by updates depending on the type of read operation and the type of update operation. For example, a watch set by getChildren is triggered after a create or delete of a child, but not by any setData on its children, as that does not change the result of getChildren. ZK-MASON notifies the client when its *watch* is triggered.

**ZK-MASON execution protocol.** ZK-MASON's execution protocol is based on Eris's execution protocol [37]. ZK-MASON assigns znodes to shards based on a hash of the full pathname. Shards consist of $2f + 1$ servers; each shard tolerates $f$ failures. Each server executes incoming operations in order of the shard's sequence space. When a proxy receives a client operation, the service stub determines which shards are involved in the operation and requests a multi-sequence number for the relevant sequence spaces. For example, to execute a create, the service stub hashes the *path* and the parent pathname to get the sequence spaces for those two shards. MASON acquires and replicates a multi-sequence number with the two sequence spaces. The service stub sends a create operation to each server in *path*'s shard and an addChild operation to each server in *path*'s parent's shard in parallel. When the stub receives a quorum of $f + 1$ responses from each shard, the operation is complete; the stub informs MASON of completion, and MASON returns to the client. Read operations, for example, getData, only need to receive one response from the shard before returning to the client because they are sequenced and do not change state.

**Ephemeral znodes.** *Ephemeral znodes* are transient znodes that exist only during an active client connection. They are created by a client and deleted by the service when the client disconnects, either explicitly or due to failure. Ephemeral znodes can be used to add to a distributed queue: if the creating client fails, the object is removed. They can also help manage locks: if a client acquires a lock and fails, the lock is released when the ephemeral object disappears [53]. Implementing ephemeral znodes in ZK-MASON is straightforward. Shards keep a timer that is reset with client heartbeats. After timing out, the shard sends a delete to a proxy to delete the node. The delete is ordered to prevent divergent shards.

**The contiguous multi-sequence abstraction simplifies ZK-MASON.** Implementing this service over a noncontiguous multi-sequence would require consensus to deal with

holes. Because a missing sequence number could belong to a multi-shard operation, e.g., create, the hole-filling consensus would need to be service-wide to avoid partially executing the operation on some shards but not others. To handle cases where aborting a partially-executed operation is impossible, each full operation would need to be persisted by the service so it could be recovered by shards that never received it (e.g., the full operation could be sent to every relevant shard).

In ZK-MASON, if a shard encounters a gap in its sequence space, it can wait for the missing operation and each shard only needs to receive the parts of the operation that will execute on that shard. The contiguous multi-sequence guarantees that the operation will be executed.

## 7 Evaluation

MASON provides two main innovations for building services. First, it is a general, reusable building block that offers the contiguous multi-sequence abstraction. This makes it easy to build efficient implementations of complex services (§6). But, as with any such abstraction, we expect overheads compared to specialized implementations. Second, MASON provides a scalable multi-sequence allowing previously unscalable services to now scale. This section quantifies the overhead of MASON's general abstraction for two services (§7.2 and §7.3), shows MASON provides scalable ordering (§7.1), that its scalable ordering does indeed enable services to scale (§7.2 and §7.3), and that MASON does provide a contiguous multi-sequence despite failures (§7.4).

**Implementation.** MASON is written in C++. All components, including clients, service shards, and MASON components, communicate with eRPC, a reliable RPC framework [27]. eRPC uses unreliable datagrams in Intel DPDK (v. 17.11.5) as its transport layer [14]. We replicate proxies with Raft [46], and periodically durably snapshot their state for Raft log compaction. We do not implement reconfiguration. MASON's source code is available at https://github.com/princeton-sns/mason.

**Evaluation setup.** We evaluate MASON on the Emulab testbed [57] with Dell R430 (d430) machines [11]. We run Ubuntu 18.04.11 with Linux kernel version 4.15.0. The machines have two hyperthreaded 8-core CPUs (Intel E5-2630 "Haswell", 2.4 GHz) with 20 MB L3 cache, 64 GB RAM, and one dual-port 10 GbE PCI-Express NIC (Intel X710).

We load MASON with clients running on separate machines of the same type. Unless otherwise specified, each client machine runs 16 threads, each implementing several logical closed-loop clients that generate new operations as previous operations complete. We control load by varying the number of client machines and the number of logical closed-loop clients per thread. Latency is measured at clients for each operation. We report the median over five trials of the median latency over all clients in a trial. We present latency as *median/99th percentile*. Throughput is also measured at each

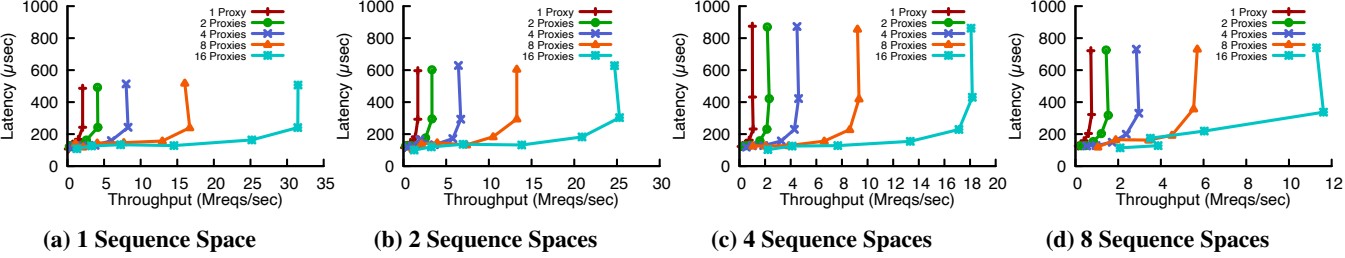

**Figure 3:** MASON ordering throughput-latency; each point represents a given load, doubling the client load from the previous point. MASON scales linearly with the number of proxies: as the number of proxies doubles, the ordering throughput also roughly doubles for each sequence space count.

client and aggregated over all clients in a trial. For all scalability experiments we derive the throughput by increasing load (i.e., the number of logical clients). We report the highest throughput before latency spikes from overload. We show the median throughput over five trials. Trials are 68 seconds each; the first and last 4 seconds of measurements are discarded.

Each proxy is replicated on 3 machines. Experiments in Sections 7.1 and 7.4 use a stub service with one operation: clients indicate relevant sequence spaces and the service returns the assigned multi-sequence number to the client.

## 7.1 MASON Scales Ordering Throughput

MASON uses two mechanisms to scale ordering throughput: adding more proxies and increasing batching to the sequencer. The first mechanism, adding more proxies, is evaluated in Figure 3. Ordering throughput is the number of client operations per second that receive a multi-sequence number and return to clients. To stress ordering throughput, the proxies do not execute operations on behalf of clients in this experiment. We present latency as *median/99th percentile*.

Figure 3 shows that, as the number of proxies doubles, the ordering throughput also roughly doubles for each sequence space count. As the number of sequence spaces in the system increases, the per-proxy machine throughput decreases, so overall ordering throughput with the same number of proxies is lower. Latency at these throughputs ranges from ~243 (median)/~380 $\mu$s (99th percentile) for a single sequence space to ~358/~693 $\mu$s for 8 sequence spaces. This experiment demonstrates that adding more proxies enables MASON to scale ordering throughput.

We are unable to test our second mechanism, increasing batching to the sequencer, because we cannot saturate the sequencer with the machines available on Emulab. With 48 proxy machines, the sequencer processes ~3.2 Mops/s, which is far from the ~14.5 Mops/s possible at line rate. As MASON scales linearly with increasing proxies, we expect to be able to achieve over 142 Mops/s before the sequencer becomes the bottleneck. At that point, we expect to be able to continue doubling the ordering throughput of MASON by doubling the number of proxies and doubling the batch sizes. Average batch size for 48 proxies with one sequence space is ~8 operations.

## 7.2 Making CORFU Scalable

MASON provides scalable ordering that, when coupled with a scalable execution protocol, enables services to scale. Corfu-MASON replaces CORFU's monolithic sequencer with MASON, yielding a scalable distributed shared log (§6.2).

We compare Corfu-MASON with CORFU′, our implementation of CORFU in the same environment as Corfu-MASON, using C++ and eRPC over DPDK. CORFU′'s sequencer processes requests at ~14.2 Mops/s, nearly line-rate for our message size (~14.5 Mops/s). This is a fairer baseline than using CORFU's improved sequencer, whose maximum ordering throughput is ~570 Kops/s [3, 4].

Figure 4a evaluates Corfu-MASON's scalability. We run a workload consisting entirely of 64 B appends and increase the number of Corfu shards. We use 6 (replicated) proxies for every Corfu shard, keeping the ratio of proxies to Corfu shards constant. CORFU′ roughly doubles throughput from one to two Corfu shards before the sequencer saturates and latency increases; the maximum observed throughput of CORFU′ is ~14.1 Mops/s with latency of ~70 (median)/~90 $\mu$s (99th percentile). MASON allows ordering in Corfu-MASON to scale, enabling service throughput to increase linearly: Corfu-MASON scales from ~7.3 Mops/s with one Corfu shard to ~29.1 Mops/s with four Corfu shards, an increase of ~3.98x. Append latency at four Corfu shards is ~200/297 $\mu$s. The increase in latency is from extra round trips (clients sending requests to proxy leaders, which leaders replicate) and proxies waiting for 20 $\mu$s to batch requests.

Figure 4b shows the scalability of reads. Clients execute reads on random log positions in CORFU′ by reading a shard's tail replica. Reads in Corfu-MASON are executed by proxy leaders, which read the tail replica. Reads are not sequenced in either service, so reads scale the same in both services. Latency for Corfu-MASON is ~97/~147 $\mu$s, ~65 $\mu$s higher than CORFU′'s ~32/~62 $\mu$s, from the extra round trip through the proxy leader.

## 7.3 Making ZooKeeper Scalable

ZK-MASON is a ZooKeeper-like coordination service [22] (see Sec. 6.3). ZK-MASON uses a scalable execution protocol with MASON's scalable ordering to scale the entire service.

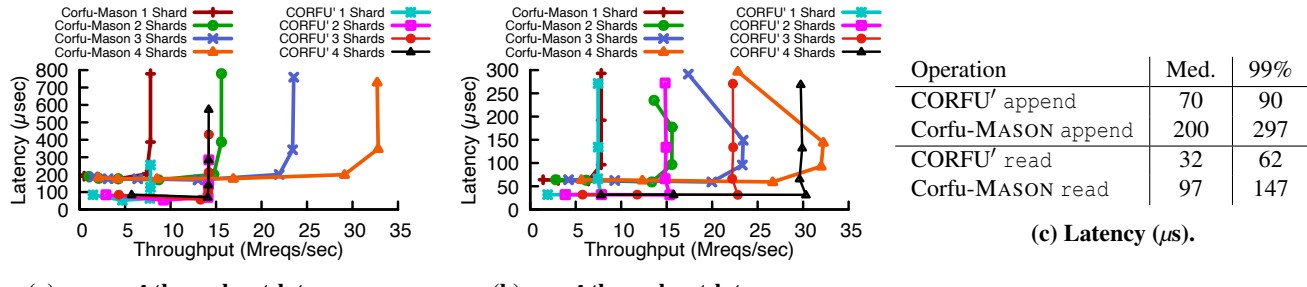

**(a)** `append` **throughput-latency**

**(b)** `read` **throughput-latency**

| Operation | Med. | 99% |
|---|---|---|
| CORFU$'$ append | 70 | 90 |
| Corfu-MASON append | 200 | 297 |
| CORFU$'$ read | 32 | 62 |
| Corfu-MASON read | 97 | 147 |

**(c) Latency ($\mu$s).**

**Figure 4: CORFU$'$ and Corfu-MASON comparison; each point represents a given load, doubling the client load from the previous point. Corfu-MASON `append` throughput scales linearly with more shards while CORFU$'$ saturates at 2 shards. Corfu-MASON has higher latency in exchange for contiguity and linear scalability.**

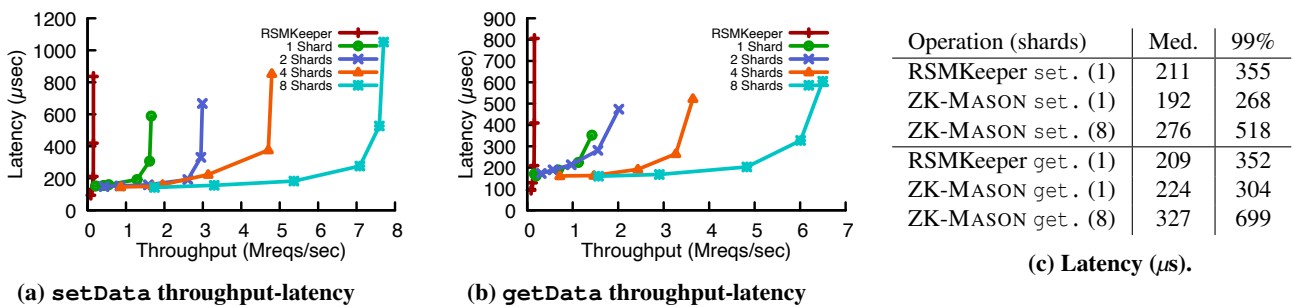

**(a)** `setData` **throughput-latency**

**(b)** `getData` **throughput-latency**

| Operation (shards) | Med. | 99% |
|---|---|---|
| RSMKeeper set. (1) | 211 | 355 |
| ZK-MASON set. (1) | 192 | 268 |
| ZK-MASON set. (8) | 276 | 518 |
| RSMKeeper get. (1) | 209 | 352 |
| ZK-MASON get. (1) | 224 | 304 |
| ZK-MASON get. (8) | 327 | 699 |

**(c) Latency ($\mu$s).**

**Figure 5: RSMKeeper and ZK-MASON comparison; each point represents a given load, doubling the client load from the previous point. ZK-MASON achieves higher throughput than RSMKeeper with a single shard at comparable latency. ZK-MASON throughput scales linearly at the cost of a modest increase in latency.**

To compare ZK-MASON and ZooKeeper we implemented RSMKeeper, a prototype of ZooKeeper over Raft [46]. RSM-Keeper has the same operations as ZK-MASON. Both are implemented in C++ with eRPC over DPDK [14, 27]; RSM-Keeper uses a single thread. We note that RSMKeeper has much higher throughput than the original ZooKeeper implementation, providing a fairer baseline.

We configured RSMKeeper and ZK-MASON to maximize service throughput while keeping latency low. RSMKeeper is loaded by one client machine running 8 threads. ZK-MASON clients use 16 threads. ZK-MASON uses 2 proxies per shard and 1 client machine per proxy. Each proxy uses 8 threads and each ZK-MASON shard uses 1 thread. This is the minimal setup for a single shard that stresses the shard's throughput. We add more ZK-MASON shards, keeping the ratio of clients and proxies to shards constant. Our ZK-MASON experiments show the scalability of the contiguous multi-sequence abstraction when scaling out the number of shards.

Figure 5a shows the throughput-latency of `setData` operations. RSMKeeper's (and ZooKeeper's) design uses a single replicated state machine to ensure consistency and thus cannot run with more than one shard; its maximum throughput is ~150 Kops/s. With one shard, ZK-MASON has $8.6\times$ the service throughput of RSMKeeper, at ~1.29 Mops/s while providing latency in a similar range as shown in Figure 5c.

ZK-MASON has lower latency than RSMKeeper in Figure 5c, ~192 $\mu$s vs ~211 $\mu$s, because of where we determined overload to be for RSMKeeper; we chose a point in the throughput-latency curve that increased throughput at the cost of some latency. At lower load and lower throughput settings, RSM-Keeper has lower latency than ZK-MASON. For example, RSMKeeper has ~94 $\mu$s median latency at ~85 Kops/s and ZK-MASON has ~152 $\mu$s at ~212 Kops/s. ZK-MASON's higher single-shard throughput comes from the proxy layer scaling with two (replicated) proxies handling client requests for one ZK-MASON shard. Furthermore, ZK-MASON shards do less work per `setData` operation than RSMKeeper. For each operation, RSMKeeper handles operation execution, one round of client-to-leader communication, two rounds of leader-to-follower communication, and snapshotting Raft state and log compaction to disk. On the other hand, MASON frees the ZK-MASON shard from handling tasks related to ordering and consensus. The shard only handles execution and one round of proxy-to-shard communication. With more resources devoted to execution, one ZK-MASON shard has a higher maximum throughput than RSMKeeper. More importantly, ZK-MASON is able to scale throughput by increasing the number of shards and proxies: with eight shards its throughput scales to ~7.1 Mops/s.

Figure 5b shows the throughput-latency of `getData` op-

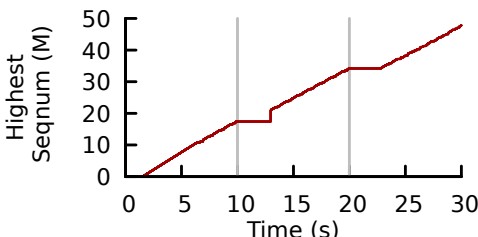

**Figure 6: Highest contiguous multi-sequence number received across all clients at time *t*. We induce proxy leader failure at 10 s and sequencer failure at 20 s.**

erations. We configured RSMKeeper to replicate `getData` operations to provide the same consistency as ZooKeeper's `sync-getData` construction and ZK-MASON's `getData` operation. RSMKeeper's maximum throughput is ~150 Kops/s with latency ~209 (median)/~352 µs (99th percentile). ZK-MASON's `getData` throughput scales from ~1.1 Mops/s with one shard to ~6 Mops/s with eight shards. Latencies in those runs range from ~224/~304 µs (one shard) to ~327/~699 µs (eight shards). `getData` operations have slightly higher latency than `setData` operations because proxies need to wait for a response from a ZK-MASON replica which must execute all operations ordered before the `getData` before returning to the client, while `setData` can be executed on ZK-MASON shards asynchronously.

### 7.4 MASON Provides a Contiguous Sequence

This experiment validates that MASON provides a contiguous sequence despite component failures. We run MASON with 16 proxies. Each proxy machine hosts either 8 leaders or 8 followers in 8 different proxies for a total of 6 proxy machines (2 leader machines and 4 follower machines). Load is generated by 4 client machines. Clients request one sequence number from each of 4 sequence spaces. We inject proxy and sequencer failure; network drops occur naturally.

Figure 6 shows the highest contiguous sequence number successfully received by a client over time for each of 4 sequence spaces. That is, if Figure 6 indicates that at time *x* the highest contiguous sequence number from a sequence space is *y*, then each sequence number up to and including *y* in that space was received by some client. We ran the experiment with 4 sequence spaces and plotted the highest contiguous sequence number for each sequence space. Since clients request one number from every sequence space, they advance at the same rate and thus all four lines overlap.

We first kill a proxy machine hosting 8 proxy leaders 10 s into the experiment. The 8 recovering proxies stop processing client operations and may have uncompleted operations. The flat region in the plot indicates where the sequence increase is blocked by uncompleted operations. Once failover is complete, the new leaders respond to pending client operations. The plot spikes as gaps in the sequence are filled in and operations serviced by the two non-failing proxies are accounted

for. Proxy failure detection and failover take 3.06 s, including 1 s-2 s for the failure detection timeout, set randomly by Raft.

We kill the sequencer 20 s into the experiment. A proxy times out 1 s later and begins the recovery protocol. Failure detection and recovery take 2.38 s—the plot's 2nd flat region—and then the contiguous multi-sequence continues to grow.

## 8 Limitations

**Resource Cost.** MASON provides a contiguous, scalable multi-sequence using replicated proxies. This ability to scale adds overhead compared to non-scalable designs in operational settings that do not demand more throughput than the latter can support. For instance, when CORFU′ saturates its sequencer at 2 Corfu shards, Corfu-MASON uses 36 proxy machines (12 proxy groups). Proxies process more RPCs than Corfu shards, so Corfu-MASON needs more proxies than shards to saturate the shards. The resource overhead (number of machines) is thus 600% to saturate two Corfu shards with Corfu-MASON (42 total machines including a standby for Corfu-MASON and 6 including a standby for CORFU). As another example, the overhead for the single-shard setup for ZK-MASON is 266% (8 total machines including a standby for ZK-MASON and 3 total machines for RSMKeeper).

However, MASON's scalability means it can be used to provide throughput beyond what can be achieved with non-scalable designs. We evaluated up to a 206% throughput increase for Corfu-MASON over CORFU′'s *maximum* throughput and expect throughput to continue scaling. We evaluated up to a 4733% throughput increase for ZK-MASON over RSMKeeper's *maximum* throughput and expect throughput to continue scaling.

Thus, a practical deployment strategy could be to initially deploy the service in a small one proxy setup and to colocate the service's processes on the machines used for the proxy. As throughput demands increase, the service could then add more proxy groups and eventually split proxies and the service's processes into different machines to scale them independently. This strategy may add some operational overhead from changing configurations but would enable a service to pay a lower cost for an initial setup.

**Performance predictability.** The intermediate components between clients and the service can make the performance of the system as a whole less predictable because tail latencies at each hop can accumulate. Furthermore, MASON's additional components may complicate performance debugging since more components will need to be inspected.

## 9 Related Work

This section explains MASON's relationship to the five categories of related work it builds upon. At a high level, the primary distinction of MASON is that it provides strict serializability, unlike atomic multicast; it is scalable, unlike state machine replication and fast ordering systems; it provides multiple sequence spaces, unlike shared logs; and its

abstraction enables more efficient, specialized service implementations than distributed databases.

**Atomic multicast.** Atomic multicast guarantees messages are delivered reliably and satisfying a total order to one or more groups of processes [8, 17, 18, 20]. Unlike the order given by a contiguous multi-sequence, the total order given by atomic multicast is not strictly serializable. Atomic multicast is thus used directly in systems to provide weaker consistency guarantees [39]. It may also be augmented to provide stronger consistency [6, 34].

**State machine replication.** There is a large body of work on state machine replication (SMR) implemented with consensus [1, 12, 15, 19, 23, 25, 28, 30–33, 40, 41, 44–46, 50, 58], which provides two properties MASON aims for: a contiguous sequence via SMR's log and fault tolerance via consensus. These protocols have a fundamental throughput ceiling, the rate a single machine can execute commands in order.

Compartmentalization is a technique to scale state machine replication [58]. Compartmentalization "involves decoupling individual bottlenecks into distinct components and scaling these components independently". How MASON scales ordering can be viewed as compartmentalization: ordering, handing out multi-sequence numbers, is explicitly separated from execution, and scaled via the proxy layer and batching.

In the compartmentalized version of Multi-Paxos, a batching layer batches client requests before sending them to the leader which orders batches in the log [58]. This technique scales ordering, but is still limited to the rate a single machine can execute commands in order, and does not easily extend to the contiguous multi-sequencing abstraction.

**Distributed shared logs.** CORFU uses a monolithic sequencer to find the tail of a distributed shared log [3]. It cannot scale beyond the throughput of the sequencer. MASON can provide a contiguous sequence to a CORFU service while scaling beyond the throughput of a monolithic sequencer, but MASON requires more resources and has higher latency.

Delos [5] unifies separate shared log or storage instances into a single virtualized shared log. It inherits the scalability limitations of its underlying systems. Scalog [13] is a distributed shared log that uses a replicated ordering mechanism to reliably totally order records in a log. Scalog increases the write throughput ceiling compared to CORFU by two orders of magnitude. It increases ordering throughput using a similar technique as MASON's proxies which batch requests across clients: storage servers collect and periodically order multiple operations at once using tiered aggregators "that relay ordering information" from the layer below it up to a replicated sequencer. Scalog and MASON both guarantee that services always see a contiguous sequence of operations, unlike CORFU. Scalog's mechanism for guaranteeing contiguity is similar to MASON's, i.e., both replicate client operations before executing them. Scalog also replicates its sequencer. However, Scalog cannot be easily extended to multi-sequencing: Scalog

orders operations using a summary of operations that arrive at individual shards. Scalog's resource overhead is lower than MASON for services where replicating an operation is the same as executing the operation (e.g., a shared log). In such services, Scalog replicates the operation on servers for contiguity, which serves to execute the operation as well. The same service over MASON must replicate the operation twice, as separate steps on distinct components: the proxy layer replicates the operation for contiguity, and execution is carried out by storing the operation on service servers. For other services where executing the operation is more than just storing its input, for example in ZK-MASON, Scalog's technique of replicating for contiguity would need to be accompanied by a separate step of executing the operation. MASON's and Scalog's overheads are thus similar for such services.

ChronoLog [29] uses physical time to order records by accounting for skew among distributed components. It reports an order of magnitude higher throughput than CORFU. Like Scalog, Delos and ChronoLog cannot be easily extended to multi-sequencing: both lack mechanisms to atomically append to multiple logs. Thus, they cannot easily be modified to support strictly serializable cross-shard operations.

Chariots [43] scales by delegating the ordering of disjoint ranges of a shared log to independent servers, providing only causal consistency [38]. FuzzyLog partially orders records in exchange for better performance [39]. MASON provides the stronger guarantee of strict serializability.

**Fast ordering systems.** State-of-the-art networks or network appliances can support high-throughput, low-latency sequencing [26, 36, 37]. Unlike MASON, these sequencers cannot scale, do not provide a contiguous sequence, and are not fault-tolerant. However, such sequencers can provide sequencing with much lower latency than MASON.

Kronos provides high-throughput happens-before ordering; services totally order operations [16]. Mostly-ordered multicast uses datacenter network properties to provide consistent multicasting except during network failures or packet loss [50]. Reliable 1Pipe, 1Pipe's strongest abstraction, provides ordered communication to receiver groups where messages eventually arrive absent failures and partitions [35]. Services detect and handle lost messages with consensus, much like services using noncontiguous multi-sequences. In contrast to these systems, MASON provides the stronger abstraction of a strictly serializable, contiguous sequence.

**Distributed databases.** FoundationDB uses a single sequence space with batching to scalably implement commit timestamps [60], but does not provide contiguity or multi-sequencing. Granola uses clock-based timestamps on clients (through "client proxies" which are similar to our service stubs in that they exist on the clients (proxies) to execute Granola (service) code) and servers and coordination among shards to determine a global transaction order [10]; the clock-based timestamps do not provide contiguity or multi-sequencing. Calvin uses a sequencing layer distributed across all servers in

the system [54]. The sequencing layer synchronously batches operations, exchanges them among replicas, then exchanges them among all servers in its copy of the database. Distributing the sequencer across servers and using large synchronous batches enables the sequencer to be more scalable than a single machine sequencer. However, as more shards are added, the all-to-all communication within a copy of the database will become a bottleneck, halting scalability. Eris [37], Calvin [54], vCorfu [56], Tango [4], and other distributed databases [2, 42, 49, 51, 59, 60] provide a higher-level abstraction than MASON. It is harder for services to build efficient, specialized implementations over the distributed database abstraction compared to the multi-sequence abstraction. For instance, ephemeral znodes (§6.3) do not fit the traditional distributed database model; a service developer would implement a new replicated component to manage client connections and explicitly delete the znode at connection termination. In contrast, implementing ephemeral znodes in ZK-MASON was straightforward.

The multi-leader approach to system design, as used in, for example, Spanner [9], uses a replica designated as the leader for each shard. A shard leader coordinates with replicas in its own shard and with leaders in other shards for operations that span multiple shards. For example, the service must implement consensus to order operations within a shard, perhaps via state-machine replication, and concurrency control, like optimistic concurrency control or two-phase locking with two-phase commit, to order operations across shards. Thus, multi-leader services are more difficult to implement than services using the multi-sequence abstraction, which orders operations within a shard by assigning a sequence space to a shard and across shards by atomically allocating a multi-sequence number. Therefore, services built on the multi-sequence abstraction do not need to implement coordination across shards for ordering; they only need to implement the service semantics.

MASON's contiguous multi-sequence abstraction is an excellent candidate for implementing distributed databases. Its contiguity would eliminate significant complexity in ported implementations of Eris and vCorfu. Similarly, its contiguity would greatly simplify developing new multi-sequence-based distributed databases. Its scalable multi-sequence would enable Eris, vCorfu, and future databases to scale far higher than the throughput ceiling of monolithic sequencers. This is an important avenue for future work.

## 10 Conclusion

The multi-sequence abstraction extends the sequence abstraction to enable consistent ordering across shards with only local ordering information. This paper proposed the contiguous multi-sequence abstraction for building consistent services. It is a stronger abstraction than the noncontiguous multi-sequence abstraction in use today, making it easier to build services with multi-sequences. We also presented MA-

SON, the first system to expose the contiguous multi-sequence abstraction and the first to provide a scalable multi-sequence. We demonstrated MASON's usefulness as a building block for scalable, consistent services by using it to enable scalability in two services that were previously fundamentally unscalable.

**Acknowledgements.** We thank the anonymous JSys reviewers for their feedback. We are grateful to Mohsin Ali, who contributed to early stages of this work. We thank Jeffrey Helt, Khiem Ngo, Zhenyu Song, Jennifer Lam, and Anja Kalaba for their help improving this work. Our experimental results were made possible by the Emulab testbed [57]. This material is based upon work supported by the National Science Foundation under Grant Nos. CNS-1910390, CNS-1564242, and CNS-1835253. Any opinions, findings, and conclusions or recommendations expressed in this material are those of the author(s) and do not necessarily reflect the views of the National Science Foundation.

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

**Algorithm 1:** Sequencer Protocol

**1** $\mathcal{S}$;                  // Set of sequence spaces
**2** $atMostOnce[]$;    // Map of (proxy, $seqReqId$) to the response
**3** $activeSequencer \leftarrow False$;
**4** **when** *the sequencer receives a message m, from proxy p* **do**
**5**    **case** $m = RequestSeqNum(seqReqId, \{count_i\}_{i=0}^{|\mathcal{S}|})$ **do**
**6**       **if** $\neg activeSequencer$ **then**
**7**          **return** *null*;
**8**       **if** $(p, seqReqId) \in atMostOnce$ **then**
**9**          **return** $atMostOnce[(p, seqReqId)], True$;
**10**      $resp \leftarrow \{\emptyset\}_{i=0}^{|\mathcal{S}|}$;
**11**      **for** $i \in \{0, ..., |\mathcal{S}|\}$ **do**
**12**         **if** $count_i \neq 0$ **then**
**13**            $resp_i \leftarrow \mathcal{S}_i$;
**14**            $\mathcal{S}_i \leftarrow \mathcal{S}_i + count_i$;
**15**      $atMostOnce[(proxyId, seqReqId)] \leftarrow resp$;
**16**      **return** $resp, False$;
**17**   **case** $m = Recover$ **do**
**18**      **for** *each proxy* **do**
**19**         **send** *GetMaxAndSeal* to each proxy
**20**      **wait** for all proxies to reply
         // Portion of recovery for contiguity is omitted.
**21**      **for** $i \in \{0, ..., |\mathcal{S}|\}$ **do**
**22**         $\mathcal{S}_i \leftarrow \max_{response \in responses} \mathcal{S}_i$ in response + 1;
**23**      $activeSequence \leftarrow True$;

---

**Algorithm 2:** Proxy State and Request Protocol

**1** $curSeqReqId \leftarrow 0$;
**2** $maxCmtdSeqReqId \leftarrow -1$; // updated in ApplyLog locally
**3** $cmtdSeqReqIds[]$;          // holds all committed SeqReqIds
**4** $maxRecvdSeqNum[]$;    // max received sequence number for each sequence space
**5** $sequencers[]$;             // array of sequencers
**6** $activeIndex \leftarrow 0$;          // index of the active sequencer
**7** **when** *proxy p receives a message m* **do**
**8**    **case** $m = ClientRequest(op)$ **do**
**9**       $retx \leftarrow True$;
**10**      $seqReqId \leftarrow curSeqReqId$;
**11**      $curSeqReqId \leftarrow curSeqReqId + 1$;
**12**      $activeSequencer \leftarrow sequencers[activeIndex]$;
**13**      $nextSequencer \leftarrow sequencers[activeIndex + 1]$;
**14**      **while** $retx$ **do**
**15**         **send** $(resp, retx) \leftarrow$ $seqnumReq(myProxyId, seqReqId, op.seqReq)$ to $activeSequencer$;
**16**         **wait** for response **or suspect** $activeSequencer$ has failed;
**17**         **if** *suspect* $activeSequencer$ *has failed* **then**
**18**            **send** *Recover* to $nextSequencer$
**19**         **wait** for response from $activeSequencer$;
**20**         **if** $sequencers[activeIndex] \neq activeSequencer$ **then**
**21**            **return**
**22**         **if** $retx$ **then**
**23**            $executeNoop(resp)$;
**24**            $seqReqId \leftarrow curSeqReqId$;
**25**            $curSeqReqId \leftarrow curSeqReqId + 1$;
**26**      $replicate(seqReqId, resp)$;
**27**      **wait** for commit;
**28**      $updateMaxRecvdSeqNum(resp)$;
**29**      $execute(op)$;   // Determined by service.
**30**      **return** to $op.client$;
**31**   **case** $m = GetMaxAndSeal$ **do**
**32**      $activeIndex \leftarrow activeIndex + 1$;
**33**      $replicate(seal)$;   // contains activeIndex
**34**      **wait** for commit;
**35**      **return** $maxRecvdSeqNum$;

---

## A    Proof of Strict Serializability

This presents a proof of the strict serializability of assignment of multi-sequence numbers to operations.

### A.1    Definitions

Strict serializability requires that there exists a legal total order of operations and that the total order reflects the real-time ordering constraints. Formally: *A complete history h satisfies* linearizability *if there exists a legal total order* $\tau$ *of* $ops(h)$ *such that* $\forall op_1, op_2 \in ops(h).op_1 <_h op_2 \Rightarrow op_1 <_\tau op_2$.

That is, if an operation *x* ends before an operation *y* begins then *x* must appear before *y* in the total order.

Let $\mathcal{S} = \{\mathcal{S}_0, \mathcal{S}_1, ...\}$ be the set of all sequence spaces. We say that two multi-sequence numbers *a* and *b* *conflict* if $\exists n \in S$ such that $a_n \neq \Delta \land b_n \neq \Delta$. Multi-sequence numbers are ordered by the partial ordering $\tau$ over all multi-sequence numbers where $a < b \iff \forall n \in S, a_n \neq \Delta \land b_n \neq \Delta \implies a_n < b_n$.

Note that this partial ordering includes the possibility of conflicting multi-sequence numbers not being ordered: i.e. where $a_i < b_i \land b_j < a_j$ for some $i, j \in S \implies a \| b$.

The goal of MASON is to provide an ordering for a service built on MASON. Thus, we prove that the partial ordering $\tau$ produced by MASON is a legal total order satisfying lineariz-

**Algorithm 3:** Proxy Leader Failover Recovery Protocol

---

**1 when** *proxy replica gains Raft leadership* **do**

**2**    $UncmtdSeqReqIds \leftarrow \{i.i \in \mathbb{Z} \wedge i \leq maxCmtdSeqReqId\}$;

**3**    $UncmtdSeqReqIds \leftarrow UncmtdSeqReqIds \setminus cmtdSeqReqIds$;

**4**    **for** *seqReqId $\in$ UncmtdSeqReqIds* **do**

**5**       **send** $(resp, retx) \leftarrow seqnumReq(myProxyId, seqReqId, 0)$;

**6**       **wait** for response;

**7**       **if** *retx* **then**

**8**          $executeNoop(resp)$;

**9**    $curSeqReqId \leftarrow maxCmtdSeqReqId + 1$;

---

ability. It is then up to the service to apply the operations in the order determined by $\tau$.

An operation is *assigned* a multi-sequence number when the Raft entry containing the operation and multi-sequence number pair is committed. Multi-sequence numbers are *allocated* by the sequencer to a request from the proxy; this does not guarantee the operation for which the proxy requested a multi-sequence number will be assigned the allocated multi-sequence number.

We allow operations to be assigned a range of sequence numbers in each sequence space. We will share notation for operations and sequence numbers where for an operation $x$, $x_n$ denotes the maximum sequence number assigned to operation $x$ in sequence space $n$. The comparison $x_n < y_n$, and $x_n \leq y_n$ compares the highest assigned sequence number for operation $x$ in sequence space $n$ and the lowest assigned sequence number for operation $y$ in sequence space $n$ That is, $x_n < y_n \iff \max_{i \in x_n} i < \min_{i \in y_n} i$ and $x_n \leq y_n \iff \max_{i \in x_n} i \leq \min_{i \in y_n} i$.

The term *proxy* indicates a replicated state machine that executes the MASON protocol detailed in Alg 2 and Alg 3. *Sequencer* denotes a machine executing the protocol detailed in Alg 1. A standby sequencer may begin executing the sequencer protocol from line 17 of Alg 1 when notified by any proxy.

## A.2 Assumptions

The model consists of a set of processes, $\mathcal{P}$, which contains clients, proxy replicas, and sequencers. Processes may fail according to the crash failure model, where processes stop executing requests, and the failure is undetectable to other processes.

We assume an asynchronous network model where messages can be arbitrarily delayed and reordered.

We develop MASON's proxies with Raft and assume the following as guarantees from Raft [46], the guarantee A.1 being explicitly stated in the paper.

**Guarantee A.1** *"If a log entry is committed in a given term, then that entry will be present in the logs of the leaders for all higher-numbered terms."*

**Guarantee A.2** *Raft is available as long as a majority of replicas have not failed.*

## A.3 Proof of total order

To prove that MASON provides a linearizable ordering we first show that its ordering is a total order and then prove that the total order respects the real-time order.

To provide a total order MASON needs to ensure for any two operations $x$ and $y$ one of $x < y$, $y < x$, or $\forall n \in S, x_n = \Delta \vee y_n = \Delta$. The latter case describes when the two operations share no sequence spaces, which we will call *strictly concurrent* and denote $x \|_s y$; in this case $x$ and $y$ are trivially ordered in either order. When any of these relations are true we will say operations $x$ and $y$ are *strictly ordered*. More specifically for any two operations $x$ and $y$, $x$ and $y$ are strictly ordered if and only if $(x_n < y_n \forall n \in S.(x_n \neq \Delta \wedge y_n \neq \Delta)) \vee (y_n < x_n \forall n \in S.(x_n \neq \Delta \wedge y_n \neq \Delta))$.

We prove that MASON provides a total order, that is, where all operations are strictly ordered as described above.

**Lemma A.1** *The assigned multi-sequence numbers for any replicated and committed operation do not change.*

*Proof:* Directly implied by SMR Guarantee A.1; any elected leader will have the operation and multi-sequence number pairing in its log. $\square$

The goal then is to prove that any two *assigned* operations are totally ordered, that is we need to show that $\forall x, y, (x < y) \vee (y < x) \vee x \|_s y$. We first prove a total order for conflicting operations.

**Lemma A.2** *Any two conflicting operations $x$, $y$ are strictly ordered.*

*Proof:* We prove by case analysis on all possible combinations of failures of MASON components.

**Case 0: No failures.** The sequencer trivially guarantees the existence of a total order in normal operation. Consider any two conflicting operations $x, y \in ops(h)$ and any two sequence spaces on which $x$ and $y$ conflict, $n, m$, such that $x_n \neq \Delta \wedge y_n \neq \Delta \wedge x_m \neq \Delta \wedge y_m \neq \Delta$. Without loss of generality let $x$ arrive at the sequencer before $y$. Let $\mathcal{S}_n = i, \mathcal{S}_m = j$ when $x$ arrives. Lines $11 - 14$ of Alg 1 increment $\mathcal{S}_n$ and $\mathcal{S}_m$ by the respective counts before responding. Once the proxy replicates the assigned multi-sequence numbers for $x$ the assignment does not change by Lemma A.1. Then, $y$, arriving later, must receive $\mathcal{S}_n \geq i + x_n.count, \mathcal{S}_m \geq i + x_m.count$ where $x_n.count$ is the number of sequence numbers requested for $\mathcal{S}_n$ by operation $x$ (line 14 of Alg 1). Thus, $x_n < y_n \wedge x_m < y_m$; that is, they are strictly ordered.

**Case 1: Proxy follower failure.** This case is equivalent to Case 0 by SMR guarantee A.2: proxies execute as normal with a majority of non-failing machines in the proxy.

**Case 2: Proxy leader failure.** Consider two conflicting operations $x, y$, and any two sequence spaces on which $x$ and $y$ conflict $n, m$. Upon proxy leader failure there are four cases.

*Case 2a: x and y are assigned (committed) before failure.* This case is equivalent to Case 0 by Lemma A.1.

*Case 2b: Neither x nor y are assigned before failure.* When $x$ and $y$ are retransmitted by their clients (not shown) they will be allocated *seqReqIds* greater than *maxCmtdSeqReqId*, by line 9 of Alg 3 and 10 of Alg 2. Without loss of generality consider $x$ and its *seqReqId*, *x.seqReqId*. If the sequencer has already allocated a multi-sequence number for *x.seqReqId* the sequencer responds with *retx == True* and the new leader will allocate a new *seqReqId*, by lines 8 – 9 of Alg 1 and lines 14 – 25 of Alg 2. The *x.seqReqId* is then incremented and the request to the sequencer is resent lines 14 – 25 of Alg 2. This is repeated, line 14 of Alg 2, until the sequencer has not allocated a multi-sequence number for *x.seqReqId*, indicated by returning *retx == False*, line 16 of Alg 1 and line 14 of Alg 2. $x$ is then allocated a new multi-sequence number. Thus, $x$ and $y$ eventually receive new sequence numbers and this case is equivalent to Case 0.

*Case 2c: Either x or y is assigned before failure, and the other is not.* Without loss of generality assume $x$ is assigned a multi-sequence number and $y$ is not. The logic is similar to Case 2b. When $y$ is retransmitted by its client (not shown) it will be allocated a *seqReqId* greater than *maxCmtdSeqReqId*, by line 9 of Alg 3 and 10 of Alg 2. Consider $y$'s *seqReqId*, *y.seqReqId*. If the sequencer has already allocated a multi-sequence number for *y.seqReqId* the sequencer responds with *retx == True* and the new leader will allocate a new *seqReqId*, by lines 8 – 9 of Alg 1 and lines 14 – 25 of Alg 2. The *y.seqReqId* is then incremented and the request to the sequencer is resent lines 14 – 25 of Alg 2. This is repeated, line 14 of Alg 2, until the sequencer has not allocated a multi-sequence number for *y.seqReqId*, indicated by returning *retx == False*, line 16 of Alg 1 and line 14 of Alg 2. $y$ is then allocated a new multi-sequence number. Thus, $y$ eventually receives a new sequence number and this case is equivalent to $y$ arriving to the sequencer later as in Case 0. These subcases exhaust all 4 combinations of the state of processing of $x$ and $y$.

**Case 3: Sequencer failure.** All multi-sequence numbers replicated (and assigned) before sequencer failure are totally ordered by Case 0 and Lemma A.1. What remains to show is that all multi-sequencers assigned after failure are totally ordered. No multi-sequence number allocated by the previous sequencer will be assigned after line 34 of Alg 2 because of lines 32 and 20 – 21 of Alg 2. Proxies trivially ensure *maxRecvdSeqNum* $\geq$ all assigned multi-sequence numbers at commit time, line 28 of Alg 2. Thus, for any assigned multi-sequence number $x$ at the time of seal commit: $x_i \leq S_i.i \in S$,

by lines 21 – 22 of Alg 1. For any multi-sequence number, $y$, assigned after recovery, $S_i < y_i.i \in S$, line 14 of Alg 1. So, $x < y$ for any pair $(x, y)$ where $x$ is assigned before recovery seal and $y$ is assigned after recovery seal. $\forall j, k \in ops(h).j, k$ assigned after recovery, $j$ and $k$ are strictly ordered or strictly concurrent by Case 0.

**Case 4: Concurrent proxy leader and proxy follower failure.** This case is equivalent to Case 2 by the guarantee of availability when fewer than a majority of machines failed A.2.

**Case 5: Concurrent proxy follower failure and sequencer failure.** As a guarantee of SMR, proxies continue to operate as normal with a majority of non-failing machines (A.2). Thus, this case is equivalent to Case 3.

**Case 6: Concurrent proxy leader and sequencer failure.**

*Case 6a: The sealing operation on the proxy was not replicated.* The new sequencer cannot execute the recovery process until it receives confirmation from every proxy that they were sealed, line 20 of Alg 1. Sealed confirmations are not sent until the seal is replicated. Thus, the new leader will eventually hear, via retransmits, from the new sequencer, and begin replicating the seal, line 33 of Alg 2. Thus, this case is equivalent to Case 3.

*Case 6b: The sealing operation on the proxy was replicated.* SMR guarantees that only a replica with all committed operations can become the new leader, guarantee A.1. Thus, the new leader has the seal operation, and begins to execute recovery. Thus, this case becomes equivalent to Case 3. These two cases are exhaustive as the proxy either committed the seal command or did not at any point in time.

**Case 7: Concurrent proxy leader, proxy follower, and sequencer failure.** This case is equivalent to Case 6 by the guarantees of SMR when $f$ or fewer replicas fail.

These cases are exhaustive because they are all combinations of possible failures of components in MASON. $\square$

**Lemma A.3** MASON*'s ordering is a total order, that is,* $\forall$ *assigned operations* $x, y, (x < y) \lor (y < x) \lor x \|_s y.$

*Proof:* Either $x$ and $y$ conflict or they do not. If $x$ and $y$ conflict, then they are totally ordered by Lemma A.2. If they are non-conflicting, then they are strictly concurrent and can be ordered by $\tau$ in any order. $\square$

## A.4 Proof of real-time order

We need to show for any operation $x$ returned to a client, any operation $y$ invoked after $x$ returned is ordered after $x$ in the total order. We denote the event of the response to a client as $resp(op)$ and the invocation event $inv(op)$.

**Lemma A.4** *If an operation, x, is assigned a multi-sequence number, n, then a sequencer allocated n for x.*

*Proof:* Lines 9 – 26 of Alg 2 imply that the proxy only replicates, assigns, an operation if *retx* is False (line 14). This

implies the returned $n$ was allocated for $x$, lines $8-16$ of Alg 1. □

**Lemma A.5** *For any two operations $x$ and $y$, $resp(x)$ precedes $inv(y)$ in real-time implies $x < y$.*

*Proof:* Given any two operations $x$ and $y$ and, without loss of generality, assume $resp(x)$ precedes $inv(y)$ in real-time, there are two cases $x$ and $y$ conflict or they do not.

**Case 0: $x$ and $y$ do not conflict.** In this case $x$ and $y$ are strictly concurrent and can be assigned in either order. We order $y$ after $x$ in the total order.

**Case 1: $x$ and $y$ conflict.** Given Lemma A.4, it is sufficient to show that for any $y$ invoked after $resp(x)$, $y$ is *allocated* a higher multi-sequence number than $x$, such that $x < y$. There are thus two cases: the sequencer that allocated the assigned multi-sequence number for $x$ allocates the assigned multi-sequence number for $y$ or it does not.

*Case 1a: The sequencer that allocated the assigned multi-sequence number for $x$ allocates the assigned multi-sequence number for $y$.* In this case $x < y$ by the normal case ordering. Specifically $resp(x) < inv(y)$ implies that $x$, being already assigned, arrives to the sequencer before $y$. Line $11-14$ of Alg 1 increases all sequence spaces for which $x$ requested a sequence number. Thus, the conflicting sequence spaces are increased. The sequence spaces on any sequencer do not decrease, thus, $y$ is allocated a higher multi-sequence number. So, $\forall n \in S.x_n \neq \Delta \wedge y_n \neq \Delta, x_n < y_n$, thus $x < y$.

*Case 1b: The sequencer that allocated the assigned multi-sequence number for $x$ does not allocate the assigned multi-sequence number for $y$.* Without loss of generality let the sequencer that allocates the multi-sequence number eventually assigned to $x$ be $S_x$ and the sequencer that allocates the multi-sequence number eventually assigned to $y$ be $S_y$. Because $y$ is assigned a multi-sequence number allocated by $S_y$ and $x$ is assigned a multi-sequence number allocated by $S_x$ and $resp(x) < inv(y)$, $S_y$ must have become the active sequencer after $x.seqnum$ was allocated. To become the active sequencer $S_y$ must have received a *Recover* message from a proxy and executed recovery, receiving the *maxRecvdSeqNum* from every proxy (lines 3, 17, $18-20$, and 23 of Alg 1 and $31-35$ of Alg 2). As $S_x$ allocated the sequence number eventually assigned to $x$, $x.seqnum$ must be *assigned* (replicated) before the proxy receives *GetMaxAndSeal* and replicates the *seal*; otherwise, the proxy would have incremented *activeIndex* and began to ignore messages from $S_x$, lines 20 $-21$ and $30-31$ of Alg 2. Thus, $x.seqnum$ is replicated before *seal* and the proxy replies with a multi-sequence number, *max* such that $\forall n.x_n \neq \Delta, x_n \leq max_n$. Thus, $S_y$ will have $x_i \leq max_i < S_i \forall i.x_i \neq \Delta$, line 19 of Alg 1 and line 35 of Alg 2. The sequence spaces on any sequencer do not decrease and so $\forall n.x_n \neq \Delta \wedge y_n \neq \Delta, x_n \leq max_n < y_n$. Thus, $x < y$. □

**Theorem A.1** MASON *provides a strictly serializable total ordering.*

*Proof:* MASON provides a total order, by A.3, that respects real-time ordering, by A.5. □

