# OpenReview forum: "Revision: Solution: Scalable, Contiguous Sequencing for Building Consistent Services"
_JSYS/2023/March_Papers — Accept (with shepherding)_

### Official Review · Reviewer_1UXG · 2023-03-22

**Decision:**

Strong accept: excellent paper that will help the community

**Strengths:**

- With the popularity of deterministic databases, fast and reliable sequencers are an important problem to solve. The idea of having a reliable sequencer that can be used as a service without worrying about sequence holes and recovery is interesting.
- The paper shows the existence of more complicated failure scenarios and possible sources of bottlenecks due to the increased number of components and addresses most of them.
- The evaluation shows remarkable throughput improvements in tested systems.
- Mason seems easy enough to use, allowing building scalable multi-shard services with relatively little overhead.
- The paper is well written. It is easy to read and follow. Overall, I was very engaged reading it.
- Proofs of provided guarantees.

**Weaknesses:**

- The paper still lacks the evaluation comparison to other sequencer solutions. While authors claim they do not scale, seeing how much MASON outperforms other sequencers is important.
- I still stand by my point of bloated resource usage of the proposed solution, although this has been made clear in the revised submission, so I will not count this against the paper.

**Detailed Comments:**

A minor point:
- "These two properties enable a leader takeover protocol that allows MASON to only replicate once, discussed below (§4.1)"
  - I am not sure I follow this sentence, especially the "only replicate once" part. Moreover, when authors talk about the "replicated proxy" for the first time (in the intro) the readers do not really know what is being replicated by these proxies. In fact, we learn that proxies replicate requests+sequence number only in section 4. Even section 3.2, which introduces the proxy component, does not discuss what exactly is being replicated, making the above sentence hard to understand.

**Expertise:**

Actively publishing in this area

**Summary Of Review:**

This paper presents MASON, a reliable sequencer layer for large distributed, sharded systems. MASON assigns a multi-sequence number to each operation, and this number determines the execution order, essentially building a deterministic (i.e., with the pre-ordered transaction) transactional system. MASON addresses several problems with prior sequencing systems. More specifically, if a system fails after acquiring the sequence number, it will create a "hole" in the ordered history. MASON avoids this problem by durably remembering each transaction/request before assigning it a sequence. For a longer, more detailed summary, see the initial review.

The revised version adds details on resource usage, failures and recovery that greatly improve the paper.

I appreciate the authors taking the time to revise and improve the paper. Most of my initial comments and concerns have been sufficiently addressed, so I am happy to accept the paper!

**Useful:**

yes

---

### Official Review · Reviewer_dZXW · 2023-04-12

**Decision:**

Strong accept: excellent paper that will help the community

**Strengths:**

I have recommended to accept the paper before with 2 suggestions, and I am very pleased to see both suggestions are addressed in this revision.

1. "a diagram that shows an example of how the failure leads to potential holes in multi-sequence space."

Authors added Figure 2 for elaboration.

2. "instead of a bar graph for throughput and table with few rows for latency, plotting the throughput v.s. latency in the same graph would be a much better choice."

Authors added the new graphs in Figure 3, 4, and 5. These graph now clearly show the improved scalability.

**Weaknesses:**

The revision does not need further improvements.

**Expertise:**

Actively publishing in this area

**Summary Of Review:**

This paper introduced MASON, a consensus-based replicated proxy service that coordinates between clients, sharding datastore and sequencer. The proxy (or proxies) as a replicated state machine which tolerates up to f failures, provided the guarantee of any sequence number assigned to an operation will not be lost, even if sequencer and client can crash at any time. In fact, the system depends on several protocols to achieve the goal. Proxy execution protocol describes the main process. Proxy leader failover protocol ensures a new leader can identify any temporary holes and recover previous sequence or get new numbers. Sequencer recovery protocol make sure that all proxies local view of the sequence space are collected and holes filled with no-ops. Finally, a garbage collection protocol among all proxy leaders helps to reduce the cost of previous recovery protocols.

**Useful:**

yes

---

### Official Review · Reviewer_kdLu · 2023-04-17

**Decision:**

Strong accept: excellent paper that will help the community

**Strengths:**

The authors addressed my concerns about the relationship with Proxy and the need for replication, along with adding a section discussing the limitations of the proposed approach. The paper is not comprehensive.

**Weaknesses:**

No weaknesses were found in the paper after revision.

**Expertise:**

Actively publishing in this area

**Summary Of Review:**

I thank the authors for addressing all my comments. I believe the paper is in a much better shape right now and deserves publication.

**Useful:**

yes

---

### Meta-Review · Area_Chair_rVd9 · 2023-04-17

**Recommendation:** Accept
**Confidence:** 5

**Metareview:**

Dear Authors,

Thank you again for submitting the revised version of your manuscript titled "Solution: Scalable, Contiguous Sequencing for Building Consistent Services.” Reviewers have agreed to accept your paper. Congratulations! Thank you for the hard work you have put on producing this version.

As per JSys rules, your paper will undergo shepherding to produce the final camera ready version. Although all reviewers showed excitement in accepting the paper, there is still a minor suggestion we would recommend to address.  One of the reviewers asked to clarify the meaning of "only replicate once," and to address that in the introduction section, the fact that a proxy is replicated doesn't clarify what is exactly replicated until later sections in the paper.

The camera ready version is due in one month, May 17, 2023. To allow timely shepherding, please submit the revised version, with annotated changes, one week before, May 10, 2023, or earlier of course.

If you have any questions or comments, please do not hesitate to reach out to us. Thank you again for submitting to JSys, and congratulations!

Lewis Tseng and Roberto Palmieri
Area chairs of JSys

---

### Decision · Program_Chairs · 2023-04-27

**Decision:**

Accept (with shepherding)

**Comment:**

Congratulations on getting your manuscript accepted!

The meta-review contains details about the reviewers' expectations for the final version. Please reach out to the Area Chairs if anything is unclear.

Best,